# An Effect of the Space Dimension of Electron Fermi Gas upon the Spin Ordering in Clusters and Nanoparticles

**Elena Orlenko [1],***  **and Fedor Orlenko [2]**

[1]  Theoretical Physics Department, Peter the Great St. Petersburg Polytechnic University, Polytechnicheskaya St. 29, 195251 St. Petersburg, Russia

[2]  Department of Physics, St. Petersburg State Chemical-Pharmaceutical Academy (SPCPA), Professor Popov St., 14, Lit. A, 197376 St. Petersburg, Russia; fadler@mail.ru

*  Correspondence: eorlenko@mail.ru

**Abstract:** Herein, the collective effects of spin polarization in a degenerate electron gas of an arbitrary space dimension are discussed. We consider these low-dimensional systems in light of potential wells (rectangular or cylindrical), and as a two- or one-dimensional oscillator system with the second (and third) spatial dimension proportional to the oscillator's length. The concept of "intermediate" sizes $\nu = 6, 5, 4$ corresponding to the quasi-low dimensions $\nu^* = 0, 1, 2$, contrary to "pure" space dimensions $\nu = 1, 2$ is introduced. A general effect of the space dimension upon the spontaneous polarization in electron Fermi gas is detected.

**Keywords:** quasi-two; quasi-one; quasi-zero dimension Fermi system; intermediate dimension of phase space; polarized state; critical parameter; a general criterion for the spontaneous polarization

## 1. Introduction

During the last decade, the ground state of a low-dimensional electron system has been the object of intensive study [1–3]. The possibility of spontaneous magnetism is a subject of theoretical discussions using different approaches based on a concept of a quasi-1D, or quasi-2D system. Commonly, we call the system dynamic one- or two-dimensional if the electron motion is free only in one (or two) spatial dimensions and their motion in the second and third (or only third) dimensions corresponds to the discrete energy states. Practically, samples with quantum scale (films and wire) [4–6] have size approximately equal to 50–100 atomic units (a.u.). The experimentally obtained quasi-one-dimensional system on the (114) surface of the bismuth semimetal has a size of about 60–100 a.u. [6] in which magnetic properties of the electron gas are maintained. Systems where metal–insulator transition is experimentally observed in the metal–oxide semiconductor, as well as in the two-dimensional gas of current carriers in the Si–Ge and GaAs–GaAlAs heterojunctions [6,7] have the same ranges (size about 60–100 a.u.) in the "third" space dimension. The experimental dependences of the conductivity of the two-dimensional gas in the applied magnetic field near the metal–insulator transition indicate the presence of spontaneous polarization in the determined ranges of parameters. Some physical systems are typically two-dimensional versions of electron gas. These are electrons confined to the interface of metal–oxide-semiconductor sandwiches [2]. They are more adequately describable by the electron gas model. There are many surprises here, so, as was shown by the experiment in [1], the dispersion of the plasmons go to zero for long wave-length in contrast to the three-dimensional situation. Using the correct Gell–Mann–Brueckner approach the author of [2] calculated the correlation energy by summing all cyclic diagrams and taking into account exchange contributions up to the

second order for both magnetically polarized electrons with interaction and for the high-density paramagnetic state. It was found that ferromagnetic is lower in energy compared to the paramagnetic state for dimensionless parameter $r_s > 2, 3$. The results obtained here are valid only for strictly 2D-electron gas. As underlined in [2], in a real system with non-zero width, the results remain qualitatively valid, but require a quantitively detailed consideration. The total energy takes into account the negative maximal exchange contribution in the case of complete (100%) polarization of spin, as compared in [3,4,8–14] with the energy of degenerate electron gas in the case of complete depolarization. Such a comparison leads to the conjecture that a state with spontaneous polarization of electrons in a quasi-two-dimensional semiconductor is formed only when the exchange energy exceeds the kinetic energy, which is possible only when the electron concentration in a quasi-two-dimensional system is σ~$7.6 \times 10^{10}$–$5 \times 10^{11}$ 1/cm$^2$ (see experimental work [12]). The value of concentration conforms with the experiments in [6,7,15,16], where the observation of spontaneous polarization in metal-oxygen-semiconductor (MOS) silicon structures for $\sigma_c = 8 \times 10^{10}$ cm$^{-2}$ was reported. High mobility-doped semiconductor heterostructures [17] provide a model system for the study of low-energy excitations of the two-dimensional interacting electron system. Well-defined excitations of the Fermi disk have been investigated at very low temperatures by intraband spectroscopy. Since the energy of these excitations is comparable to the Fermi energy (a few meV), electronic resonant Raman scattering (ERRS) [18] in the visible range is one of the most powerful methods for such a purpose. Raman spectroscopy is able to probe excitations of the unpolarized two-dimensional electron gas with non-zero momentum on a plane where many-body interactions manifest themselves [19,20]. In the integer quantum Hall regime, spin waves, inter-Landau-level magnetoplasmons, and spin-flip waves have been evidenced [21,22]. For a more detailed experimental review see [23], which also presents theoretical aspects of spin-polarized two-dimensional electron gas achieved in doped semimagnetic quantum wells. A high mobility spin-polarized two-dimensional electron gas has been obtained in dilute magnetic semiconductor heterostructures such as $Cd_{1-x}Mn_xTe/Cd_{1-y}Mg_yTe$ n-type modulation doped quantum wells [24]. It has been investigated by electron Raman spectroscopy [25–28]. The giant Zeeman effect [27] occurring in these systems allows for the creation of a highly polarized 2D electron gas in which the spin quantization occurs without direct modification of the orbital motion. Effects of magnetic ordering associated with the Coulomb exchange interaction of free electrons in a 2D Fermi system are considered in [4,29], which has shown that the paramagnetic response is substantially enhanced by Fermi liquid effects. The phase transition to a state with spontaneous polarization of spins has been described. This should occur when Heisenberg parameter J is not smaller than $\varepsilon_F/3.06$ (approximately one-third of the Fermi energy) and not larger than half the Fermi energy. Spin-polarized 2D gas of electrons in $Cd_{1-x}Mn_xTe$ has been realized for typical values $\sigma_2 = 2.5 \times 10^{11}$ cm$^{-2}$, where polarization degree $\alpha$ as was detected in [24] as close to 100%.

The space dimension is a very important factor. It determines the number of single-electron states belonging to the same energy surface in the phase space. The main parameters of the electron system (such as Fermi energy $E_F$, as well as exchange interaction energy A, and the mean energy μBH of magnetic dipole interaction with the external magnetic field) are determined namely by numbers of (in the phase space) states. These main parameters sometimes have the same order, dependent on the space dimension, which gives rise to interesting magnetic phenomena.

The above-mentioned model of the low-dimension system, or so-called quasi-two-(one)-dimensional clusters (films and wire) [8,9,30,31], which describes an electron system as a potential well (rectangular or cylindrical) is not adequate for the systems in the abovementioned experiments. In order to provide the discrete energy specter with state distances $\Delta E \sim \frac{h^2}{2ma^2} \sim E_F$, the size $a$ of a well should be the same order as the mean distance between the two electrons in the system, which means that a typical size of well should be $a \sim a_B = 1$ a.u. Contrary to this, the real depth of two-dimension samples, as mentioned above, is $a \sim 60 \div 100$ a.u. [6,7,28,29].

Reliable descriptions of a spin ordering and of magnetic properties of the trapped atoms are presented for the quasi-one- or quasi-two-dimensional Bose and Fermi condensate. Such descriptions

are also useful for the disordered cluster electron systems. We consider these systems not as potential wells (rectangular or cylindrical), but as a two- or one- dimensional oscillator system. In these cases the second (and third) spatial dimensions are proportional to the $a_{osc} = \sqrt{\frac{\hbar}{2m\omega}} \sim$ 50 a.u. oscillator's length. In this model, the oscillator's length corresponds to the real width of the sample in the experiment, whereas the frequency $\omega$ is a mean value, determined by the size of the sample. Such an electron-trapping model allows us to manipulate with the space dimension by changing the external field frequency $\omega$ in the case of cluster systems or by varying the depth $a_{osc}$ of nanowire or nanofilms used for nanosensors. One of the possible applications of the low-dimensional electron Fermi system is the so-called spin transistor [32]. The spin transistor is a magnetically-sensitive device made out of magnetic materials. It is currently a hot area of research and is an advanced version of the conventional electronic transistor. For functioning of this device, first the spins have to be injected from the source into a non-magnetic layer and then transmitted to the collector. These non-magnetic layers are also called semimetals because of their very large spin diffusion lengths. The operation of the spin transistor relies on the ability of electrons to store the information on spin polarization. The band structure of the ferromagnetic emitter is such that an electric current is carried by electrons of one spin sub-band. It is a spin polarized current and there is a current of magnetization. The emitter is viewed as a reservoir of particles (conducting electrons) that carry a charge and a magnetic moment. There is the rule that the emitter may exchange electrons with a base (a nonmagnetic layer) only if those particles have the same "spin-polarization" as the base.

The main goal of our theoretical studies is to detect the general effect of the spatial dimension on the spontaneous polarization in an electron Fermi gas as a possible sensory mechanism. Here we introduce the concept of the "intermediate" sizes $\nu$ = 6, 5, 4 corresponding to the quasi-low space dimensions $\nu^*$ = 0, 1, 2 contrary to the "pure" space dimensions $\nu$ = 1, 2. We describe thermodynamic properties of an electron Fermi gas for intermediate space dimensions in a general form and develop a general criterion for the spontaneous polarization in electron Fermi gas.

## 2. An Effect of Space Dimension upon the Thermodynamics of the Trapped Electron Fermi Gas

Let us consider an arbitrary space dimension in the oscillator "trap" model. The energy of free trapping electrons is:

$$\varepsilon = \sum_\alpha \frac{p_\alpha^2}{2m} + \sum_\beta \frac{m\omega_\beta^2 q_\beta^2}{2} = \sum_\alpha \frac{P_\alpha^2}{2} + \sum_\beta \frac{Q_\beta^2}{2} = \sum_\nu \frac{X_\nu^2}{2}, \tag{1}$$

where a general momentum $P_\alpha = \frac{p_\alpha}{\sqrt{m}}$ and a general coordinate $Q_\beta = q_\beta \omega_\beta \sqrt{m}$. Here, the space dimension $\nu = (\alpha + \beta) = 1, 2, 3$ corresponds to the real dimensions with $\alpha = 1, (P_x,), \alpha = 1, 2, (P_x$ and $P_y)$, or $\alpha = 1, 2, 3, (P_x, P_y, P_z)$ and $\beta = 0$, determining a free electron gas. The space dimensions $\nu = (\alpha + \beta)$ = 4, 5, 6 correspond to the $\alpha = 3 (P_x, P_y, P_z)$ and $\beta = 1, (Q_x)$ that results in $\nu = 4$; $\alpha = 3 (P_x, P_y, P_z)$ and $\beta = 1, 2, (Q_x$ and $Q_y)$, that leads to $\nu = 5$; or $\alpha = 3 (P_x, P_y, P_z)$ and $\beta = 1, 2, 3 (Q_x, Q_y, Q_z)$ that results in $\nu = 6$. It determines a quasi-2 (with $\nu = 4 \propto \nu^* = 2$), quasi-1 (with $\nu = 5 \propto \nu^* = 1$), or quasi-0 (with $\nu = 6 \propto \nu^* = 0$) electron system. Because of the orthogonality and independence of all $\nu$ variables $P_\alpha, Q_\beta$ that determine a phase configuration, we may use a universal variable set $\{X_\nu\}$ instead of $\{P_\alpha, Q_\beta\}$ variables. Then, we consider the Equation (1) as an equation of a spherical surface in the $\nu$-dimensional phase space:

$$\varepsilon = \sum_\nu \frac{X_\nu^2}{2} \tag{2}$$

with a radius $R = \sqrt{2\varepsilon}$ of this hypersphere. The surface area of the $\nu$-dimensional sphere and its volume are:

$$S_\nu\left(\sqrt{2\varepsilon}\right) = \frac{2\pi^{\nu/2}}{\Gamma(\nu/2)}\left(\sqrt{2\varepsilon}\right)^{\nu-1}$$
$$V_\nu\left(\sqrt{2\varepsilon}\right) = \frac{\pi^{\nu/2}}{\Gamma(1+\nu/2)}\left(\sqrt{2\varepsilon}\right)^{\nu} \tag{3}$$

where $\Gamma(\nu/2)$ is the gamma function. A system under consideration is not a two-, one-, or zero-dimensional in the direct meaning, because the wavefunction of carriers is a function of three coordinates, and electromagnetic fields propagate in three-dimensional space. In this case, the phase space for the one-particle system is always six-dimensional. It is important to underline it in writing a normalization condition for the N-particles Fermi system:

$$
\begin{aligned}
N &= \iint g_s n_F(\varepsilon) \frac{d^3 p d^3 r}{(2\pi h)^3} = L^{6-\nu} \frac{g_s m^{(6-\nu)/2}}{(2\pi h)^3 \omega^{\nu-3}} \int n_F(\varepsilon) d^\nu X \\
&= \frac{g_s m^{(6-\nu)/2}}{(2\pi h)^3 \omega^{\nu-3}} L^{6-\nu} \int_0^\infty n_F(\varepsilon) S_\nu\left(\sqrt{2\varepsilon}\right) d\left(\sqrt{2\varepsilon}\right),
\end{aligned}
\tag{4}
$$

where $n_F(\varepsilon)$ is the Fermi–Dirac distribution; $\mu$ is the chemical potential; $T$ is temperature in the energy scale; and $g_s$ is a spin degeneration factor. A direct calculation of the integral (4) results in the following expression:

$$
\frac{N}{L^{6-\nu}} = \sigma_{6-\nu} = \frac{g_s m^{3-\nu/2} (2\pi)^{\nu/2}}{(2\pi h)^3 \omega^{\nu-3} \Gamma(\nu/2)} \left[ \frac{2}{\nu} \mu^{\nu/2} + \frac{\pi^2 T^2}{6} \left( \frac{\nu}{2} - 1 \right) \mu^{-2+\nu/2} \right],
\tag{5}
$$

which determines the Fermi energy $\varepsilon_F$ of the $\nu$-dimensional system, if the temperature $T \to 0$ is applied:

$$
\left[ \frac{(2\pi h)^3 \omega^{\nu-3} \Gamma(1+\nu/2)}{g_s m^{3-\nu/2} (2\pi)^{\nu/2}} \sigma_{6-\nu} \right]^{2/\nu} = \mu_{T=0} = \varepsilon_F,
\tag{6}
$$

Taking into account the following recurrent relation for the $\nu$-dimension concentration $\sigma_{6-\nu}$:

$$
\sigma_{6-\nu} = \left( 4\pi a_{osc}^2 \right)^{\nu-3} \sigma_\nu = \left( 4\pi \frac{h}{2m\omega} \right)^{\nu-3} \sigma_\nu,
$$

$$
\begin{aligned}
\varepsilon_F &= \left[ \frac{(2\pi h)^3 \omega^{\nu-3} \Gamma(1+\nu/2)}{g_s m^{3-\nu/2} (2\pi)^{\nu/2}} \left( 4\pi \frac{h}{2m\omega} \right)^{\nu-3} \sigma_\nu \right]^{2/\nu} \\
&= \frac{h^2}{m} (2\pi) \left( \frac{\Gamma(1+\nu/2)}{g_s} \sigma_\nu \right)^{2/\nu}
\end{aligned}
\tag{7}
$$

we find the Fermi Energy and the temperature corrections that determine the chemical potential of the $\nu$-dimensional system at the finite temperature:

$$
\frac{\Delta \mu}{\varepsilon_F} = -\left( \frac{\nu}{2} - 1 \right) \frac{\pi^2}{6} \left( \frac{T}{\varepsilon_F} \right)^2
\tag{8}
$$

We have represented the expressions of the Fermi energies and their corrections for all space dimensions in Table 1, where D* means spatial dimension of the system correspondent to the $\nu$*. We have used a Fermi integral to obtain all expressions of energy corrections, except for the two-dimensional case. We exactly developed the correction for the two-dimensional case.

**Table 1.** Fermi energies and their corrections for different space dimensions.

| | | |
|---|---|---|
| 1D $\nu = 1$ | $\varepsilon_F = \frac{h^2}{m}(2\pi)\left(\frac{\Gamma(1+1/2)}{g_s}\sigma_1\right)^{2/1}$ $\varepsilon_F = \frac{h^2}{2m}\left(\frac{\pi\sigma_1}{g_s}\right)^2$ | $\frac{\Delta\mu}{\varepsilon_F} = +\frac{\pi^2}{12}\left(\frac{T}{\varepsilon_F}\right)^2$ |
| 2D $\nu = 2$ | $\varepsilon_F = \frac{h^2}{m}(2\pi)\left(\frac{\Gamma(1+2/2)}{g_s}\sigma_2\right)^{2/2}$ $\varepsilon_F = \frac{h^2}{m}(2\pi)\left(\frac{\sigma_2}{g_s}\right)$ | $\frac{\Delta\mu}{\varepsilon_F} = 0$ $\left(\Delta\mu = T\ln\left(1 - e^{-\frac{\varepsilon_F}{T}}\right)\right.$ $\left.\approx -Te^{-\frac{\varepsilon_F}{T}}\right)$ |
| 3D $\nu = 3$ | $\varepsilon_F = \frac{h^2}{m}(2\pi)\left(\frac{\Gamma(1+3/2)}{g_s}\sigma_3\right)^{2/3}$ $= \frac{h^2}{2m}\left(\frac{6\pi^2}{g_s}\sigma_3\right)^{2/3}$ | $\frac{\Delta\mu}{\varepsilon_F} = -\frac{\pi^2}{12}\left(\frac{T}{\varepsilon_F}\right)^2$ |
| 2D* $\nu = 4$ | $\varepsilon_F = \frac{h^2}{m}(2\pi)\left(\frac{\Gamma(1+4/2)}{g_s}\sigma_4\right)^{2/4}$ $= \frac{h^2}{ma_{osc}}\left(\frac{2\pi\sigma_2}{g_s}\right)^{1/2}$ $\sigma_{6-4} = \left(4\pi a_{osc}^2\right)^{4-3}\sigma_4$ | $\frac{\Delta\mu}{\varepsilon_F} = -\frac{\pi^2}{6}\left(\frac{T}{\varepsilon_F}\right)^2$ |
| 1D* $\nu = 5$ | $\varepsilon_F = \frac{h^2}{m}(2\pi)\left(\frac{\Gamma(1+5/2)}{g_s}\sigma_5\right)^{2/5}$ $= \frac{h^2}{2m}\left(\frac{15\cdot\pi}{4g_s a_{osc}^4}\sigma_1\right)^{2/5}$ $\sigma_{6-5} = \left(4\pi a_{osc}^2\right)^{5-3}\sigma_5$ | $\frac{\Delta\mu}{\varepsilon_F} = -\frac{\pi^2}{4}\left(\frac{T}{\varepsilon_F}\right)^2$ |
| 0D* $\nu = 6$ | $\varepsilon_F = \frac{h^2}{2ma_{osc}^2}\left(\frac{6N}{g_s}\right)^{1/3} = h\omega\cdot\left(\frac{6N}{g_s}\right)^{1/3}$ $\sigma_{6-6} = \left(4\pi a_{osc}^2\right)^{6-3}\sigma_6$ | $\frac{\Delta\mu}{\varepsilon_F} = +\frac{\pi^2}{6}\left(\frac{T}{\varepsilon_F}\right)^2$ |

The kinetic energy *K* of proper Fermi gas with the temperature corrections taken into account determines the internal energy *U* of the ν-dimensional system:

$$U = K = \frac{(2\pi)^{\frac{\nu}{2}}m^{3-\frac{\nu}{2}}l^{6-\nu}}{\Gamma\left(\frac{\nu}{2}\right)(2\pi h)^3\omega^{\nu-3}}g_s\int\limits_0^\infty \varepsilon^{\nu/2}n_F(\varepsilon)d\varepsilon$$
$$= \frac{(2\pi)^{\frac{\nu}{2}}m^{3-\frac{\nu}{2}}l^{6-\nu}}{\Gamma\left(\frac{\nu}{2}+1\right)(2\pi h)^3\omega^{\nu-3}}g_s\left\{\mu^{\frac{\nu}{2}+1} + \frac{\pi^2 T^2}{6}\left(\frac{\nu}{2}+1\right)\frac{\nu}{2}\mu^{\frac{\nu}{2}-1}\right\}$$

Taking into account the general Equations (6) and (7) for the Fermi Energy and its correction we have:

$$U = \frac{\nu}{\nu+2}N\varepsilon_F\left\{1 + \frac{\pi^2 T^2}{6\varepsilon_F^2}\left(\frac{\nu}{2}+1\right)\right\} \tag{9}$$

The expression for Grand potential Ω (or Landau Potential) leads us to the thermodynamics state equation for the ν-dimensional electron Fermi system:

$$\Omega = -T\frac{(2\pi)^{\frac{\nu}{2}}m^{3-\frac{\nu}{2}}l^{6-\nu}}{\Gamma\left(\frac{\nu}{2}\right)(2\pi h)^3\omega^{\nu-3}}g_s\int\limits_0^\infty \varepsilon^{\frac{\nu}{2}-1}\ln\left(1 + e^{\frac{\mu-\varepsilon}{T}}\right)d\varepsilon = -\frac{2}{\nu}U. \tag{10}$$

Then, we have expressions for the entropy and for the specific heat per particle, respectively:

$$S = \frac{\nu}{6}N\frac{\pi^2 T}{\varepsilon_F},$$
$$c = \frac{C}{N} = \frac{T}{N}\left(\frac{\partial S}{\partial T}\right)_{V_\nu} = \frac{\nu}{6}\frac{\pi^2 T}{\varepsilon_F}. \tag{11}$$

These patterns and thermodynamic relations are dependent on the space dimension of the electron Fermi system. Their influence on the character of the spin ordering in such a system plays a crucial role. Before we considered the polarization effects in the Fermi system, we present our results in

the more appropriate form using the dimensionless parameter $r_s$ (see [2,4,5]). Let us introduce this dimensionless parameter as a $\nu$-dimensional concentration of particles, messed in the atomic units:

$$\frac{N}{L^\nu} = \sigma_\nu = \frac{\Gamma\left(\frac{\nu}{2}+1\right)}{r_s^\nu \cdot a_B^\nu \cdot \pi^{\frac{\nu}{2}}},$$

$$r_s = \frac{1}{a_B} \cdot \left(\frac{\Gamma\left(\frac{\nu}{2}+1\right)}{\sigma_\nu \cdot \pi^{\frac{\nu}{2}}}\right)^{\frac{1}{\nu}}, \tag{12}$$

where we used the expression of the volume of the $\nu$-dimensional sphere of Equation (3), and $N$ is the total number of electrons in the system under consideration. The current citing of Equation (12) is shown in Table 2.

**Table 2.** Dimensionless universal parameter.

| 1D $\nu = 1$ | $r_s = \dfrac{\Gamma\left(\frac{3}{2}\right)}{a_B\sqrt{\pi}\cdot\sigma_1} = \dfrac{1}{2a_B\cdot\sigma_1}$ <br> $\dfrac{N}{L^1} = \sigma_1$ |
|---|---|
| 2D $\nu = 2$ | $r_s = \dfrac{1}{a_B} \cdot \sqrt{\dfrac{\Gamma(2)}{\pi\cdot\sigma_2}} = \dfrac{1}{a_B} \cdot \dfrac{1}{\sqrt{\pi\cdot\sigma_2}}$ <br> $\dfrac{N}{L^2} = \sigma_2$ |
| 3D $\nu = 3$ | $r_s = \dfrac{1}{a_B}\left(\dfrac{\Gamma\left(\frac{3}{2}+1\right)}{\sigma_3\cdot\pi^{\frac{3}{2}}}\right)^{\frac{1}{3}} = \dfrac{1}{a_B}\left(\dfrac{3}{4\pi\cdot\sigma_3}\right)^{\frac{1}{3}},$ <br> $\dfrac{N}{L^3} = \sigma_3$ |
| 2D* $\nu = 4$ | $r_s = \dfrac{1}{a_B} \cdot \left(\dfrac{\Gamma(3)}{\sigma_4\cdot\pi^2}\right)^{\frac{1}{4}} = \dfrac{1}{a_B} \cdot \left(\dfrac{8a_{osc}^2}{\sigma_2\cdot\pi}\right)^{\frac{1}{4}},$ <br> $\dfrac{N}{L^4} = \sigma_4 = \left(4\pi a_{osc}^2\right)^{-1}\sigma_2$ |
| 1D* $\nu = 5$ | $r_s = \dfrac{1}{a_B} \cdot \left(\dfrac{\Gamma\left(\frac{5}{2}+1\right)}{\sigma_5\cdot\pi^{\frac{5}{2}}}\right)^{\frac{1}{5}} = \dfrac{1}{a_B} \cdot \left(\dfrac{\Gamma\left(\frac{7}{2}\right)\left(4\pi a_{osc}^2\right)^2}{\sigma_1\cdot\pi^{\frac{5}{2}}}\right)^{\frac{1}{5}}$ <br> $= \dfrac{a_{osc}}{a_B} \cdot \left(\dfrac{30}{a_{osc}\cdot\sigma_1}\right)^{\frac{1}{5}},$ <br> $\dfrac{N}{L^5} = \sigma_5 = \left(4\pi a_{osc}^2\right)^{-2}\sigma_1$ |
| 0D* $\nu = 6$ | $r_s = \dfrac{1}{a_B} \cdot \left(\dfrac{\Gamma\left(\frac{6}{2}+1\right)}{\sigma_6\cdot\pi^{\frac{6}{2}}}\right)^{\frac{1}{6}} = \dfrac{1}{a_B} \cdot \left(\dfrac{6}{\sigma_6\cdot\pi^3}\right)^{\frac{1}{6}} = \dfrac{2a_{osc}}{a_B} \cdot \left(\dfrac{6}{N}\right)^{\frac{1}{6}},$ <br> $\dfrac{N}{L^6} = \sigma_6 = \left(4\pi a_{osc}^2\right)^{-3}\sigma_0 = \dfrac{N}{\left(4\pi a_{osc}^2\right)^3}$ |

Here we express the Fermi energy $\varepsilon_F$ of (7), and the internal energy $U$ of the proper Fermi gas (the kinetic energy $K$) of (9), using this dimensionless parameter $r_s$:

$$\varepsilon_F = \frac{h^2}{m}\frac{2}{r_s^2\cdot a_B^2}\left(\frac{\Gamma^2\left(\frac{\nu}{2}+1\right)}{g_s}\right)^{2/\nu} = \frac{2e^2}{r_s^2\cdot a_B}\left(\frac{\Gamma^2\left(\frac{\nu}{2}+1\right)}{g_s}\right)^{2/\nu},$$

$$U = \frac{\nu}{\nu+2}N\frac{2e^2}{r_s^2\cdot a_B}\left(\frac{\Gamma^2\left(\frac{\nu}{2}+1\right)}{g_s}\right)^{2/\nu}\left\{1 + \frac{\pi^2 T^2}{6\varepsilon_F^2}\left(\frac{\nu}{2}+1\right)\right\}. \tag{13}$$

Such a representation of these main parameters will be useful for consideration of polarized states of the electron Fermi gas accounting for particle interaction.

## 3. Polarized State

The establishment of an equilibrium spin polarization is a result of the competition of two main contributions: (i) Non-force exchange, which is a consequence of the Pauli exclusive principle, due to which the kinetic energy of a Fermi gas increase when spins are unpaired because free particles should occupy higher single-particle states; (ii) a Coulomb exchange interaction directly causing a decrease of unpaired spins energy.

A total Hamiltonian of the fermion system, taking into account Coulomb interaction among the electrons, has the following form in the occupation numbers representation:

$$\hat{H} = \sum_{k,\sigma} \frac{(hk)^2}{2m} a_{k\sigma}^\dagger a_{k\sigma} + \frac{1}{2} \sum_{\substack{k\sigma \\ k'\sigma'}} \sum_q V(q) a_{k+q,\sigma}^\dagger a_{k'-q,\sigma'}^\dagger a_{k'\sigma'} a_{k\sigma},\tag{14}$$

where $\vec{k}$ is a wave vector of the free electron in the $\nu$-dimensional phase space; $\sigma$ is a spin projection on to the quantization axis; and $V(q)$ is a Fourier transform of the Coulomb interaction between the electrons.

Given that the polarized state of the fermion system with a polarization $\alpha$ degree, then:

$$\begin{aligned} N^+ - N^- &= \alpha N \\ N^+ + N^- &= N, \end{aligned}\tag{15}$$

where $N^\pm = \frac{N}{2}(1 \pm \alpha)$ is the number of up and down spins, respectively and $N$ is the total number of electrons in the system. The presence of the polarization in the system of electrons changes the position of the Fermi level in Equation (7) to:

$$\varepsilon_F^\pm = \varepsilon_F(1 \pm \alpha)^{2/\nu} = \frac{h^2 k_F^2(\alpha)}{2m},\tag{16}$$

where the Fermi wave vector $k_F(\alpha)$ value, accounting for the polarization degree has the form:

$$k_F(\alpha) = k_F(1 \pm \alpha)^{1/\nu} = 2\sqrt{\pi}[\Gamma(1 + \nu/2)\sigma_\nu]^{1/\nu}(1 \pm \alpha)^{1/\nu}.\tag{17}$$

The ground state vector of the non-interacting electron system is:

$$|\Psi_0(\alpha)\rangle = \sum_{\substack{k \le k_F(\alpha) \\ s}} a_{ks}^\dagger |0\rangle.\tag{18}$$

In this case, the mean-value of the kinetic energy per particle of the ground state with respect to polarization has the following expression:

$$\frac{K}{N} = \varepsilon_F \frac{1}{g_s} \frac{\nu}{\nu+2} \left\{ (1+\alpha)^{\frac{\nu+2}{\nu}} + (1-\alpha)^{\frac{\nu+2}{\nu}} \right\},\tag{19}$$

Using Equation (13) with the temperature corrections through the dimensionless parameter $r_s$, the mean-value of the kinetic energy has the form:

$$\frac{K}{N} = \frac{\nu}{\nu+2} \frac{2e^2}{r_s^2 \cdot a_B} \left( \frac{\Gamma^2(\frac{\nu}{2}+1)}{g_s} \right)^{2/\nu} \times \left\{ (1+\alpha)^{\frac{\nu+2}{\nu}} + (1-\alpha)^{\frac{\nu+2}{\nu}} \right\} \left\{ 1 + \frac{\pi^2 T^2}{6\varepsilon_F^2} \left( \frac{\nu}{2}+1 \right) \right\}$$

The first correction of Hartree-Fock energy is:

$$E^{(1)} = \frac{1}{2} \sum_{\substack{ks \le k_F(\alpha) \\ k's' \le k_F(\alpha)}} \sum_q \langle k+q, k'-q|V(q)|k',k\rangle - \frac{1}{2} \sum_{\substack{ks \le k_F(\alpha) \\ k's' \le k_F(\alpha)}} \sum_q \langle k+q, k'-q|V(q)|k,k'\rangle,\tag{20}$$

where the first term corresponds to the direct Coulomb interaction, whereas the second is the exchange contribution. Generally, we compute the exchange contribution in the above-mentioned oscillator trapping model, where the oscillator's wave function is taken in the VKB-form:

$$\psi(x) = \sqrt{\frac{2m\omega}{\pi p}} \sin\left\{\frac{1}{h}\int_{-a_{osc}}^{x} p\,dx + \frac{\pi}{4}\right\},$$

$$\oint p\,dx = 2\pi\frac{m\omega a_{osc}^2}{2}.$$

The result of the analytical computation is shown in Table 3 for each space dimension.

**Table 3.** Exchange mean energy.

| | |
|---|---|
| 1D $\nu = 1$ | $\frac{E_{exc}^{\pm}}{N} = -\frac{e^2 k_F^2}{\sigma_1}\left(\frac{5}{2} - 3\gamma\right)\left((1+\alpha)^2 + (1-\alpha)^2\right)$ $= -\frac{C_{1\nu=1}e^2}{r_s a_B}\left((1+\alpha)^{\frac{1+1}{1}} + (1-\alpha)^{\frac{1+1}{1}}\right)$ $C_{1\nu=1} = \left(\left(\frac{5}{2} - 3\gamma\right)\pi^2/2\right)$ |
| 2D $\nu = 2$ | $\frac{E_{exc}^{\pm}}{N} = -\frac{e^2 k_F^3}{6\pi\sigma_2}\left((1+\alpha)^{3/2} + (1-\alpha)^{3/2}\right)$ $= -\frac{C_{1\nu=2}}{r_s}\frac{e^2}{a_B}\left((1+\alpha)^{\frac{2+1}{2}} + (1-\alpha)^{\frac{2+1}{2}}\right),$ $C_{1\nu=2} = \left(4\sqrt{2}/3\pi\right)$ |
| 3D $\nu = 3$ | $\frac{E_{exc}^{\pm}}{N} = -\frac{3}{2}\frac{e^2 k_F^4}{\pi\sigma_3}\left((1\pm\alpha)^{4/3} + (1\pm\alpha)^{4/3}\right)$ $= -\frac{C_{1\nu=3}}{r_s}\frac{e^2}{a_B}\left((1+\alpha)^{\frac{3+1}{3}} + (1-\alpha)^{\frac{3+1}{3}}\right);$ $C_{1\nu=3} = 0.458$ |
| 2D* $\nu = 4$ | $\frac{E_{exc}^{\pm}}{N} = -\frac{2}{5}\frac{e^2 k_F^5 a_{osc}^2}{\pi^2\sigma_2}\left((1\pm\alpha)^{5/4} + (1\pm\alpha)^{5/4}\right),$ $\frac{E_{exc}}{N} = -\frac{C_{1\nu=4}}{r_s}\frac{e^2}{a_B}\{(1+\alpha)^{\frac{4+1}{4}} + (1-\alpha)^{\frac{4+1}{4}}\}$ $C_{1\nu=4} = \frac{2}{\pi^{1/4}}$ |
| 1D* $\nu = 5$ | $\frac{E_{exc}^{\pm}}{N} = -\frac{e^2 k_F^6 a_{osc}^5}{\pi^{5/2}\sigma_1}\left((1+\alpha)^{6/5} + (1-\alpha)^{6/5}\right),$ $\frac{E_{exc}}{N} = -\frac{C_{1\nu=5}}{r_s}\frac{e^2}{a_B}\{(1+\alpha)^{\frac{5+1}{5}} + (1-\alpha)^{\frac{5+1}{5}}\}$ $C_{1\nu=5} \approx \frac{225}{4} \approx 56$ |
| 0D* $\nu = 6$ | $\frac{E_{exc}^{\pm}}{N} = -\frac{e^2 k_F^7 a_{osc}^6}{\pi^3 N}\left((1+\alpha)^{7/6} + (1-\alpha)^{7/6}\right),$ $\frac{E_{exc}}{N} = -\frac{C_{1\nu=6}}{r_s}\frac{e^2}{a_B}\{(1+\alpha)^{\frac{6+1}{6}} + (1-\alpha)^{\frac{6+1}{6}}\},$ $C_{1\nu=6} = \frac{3}{\sqrt{\pi}}$ |

We paid attention to the regularity in the values of the power function exponents that allow us to represent the exchange Hartree-Fock mean-value energy as follows:

$$\frac{E_{exc}^1}{N} = -\frac{C_{1\nu}}{r_s}\frac{e^2}{a_B}\left((1+\alpha)^{\frac{\nu+1}{\nu}} + (1-\alpha)^{\frac{\nu+1}{\nu}}\right), \tag{21}$$

where the coefficients $C_{1\nu}$ for each space dimension are presented in Table 3. The detailed computation of the exchange contributions for the different space dimensions can be seen in [2,4,5]. Then, the total mean value of the energy per electron has the following general form in the atomic units accounting for the space dimension $\nu$:

$$\frac{E}{N} = \frac{C_{2\nu}}{r_s^2}\left((1+\alpha)^{\frac{\nu+2}{\nu}} + (1-\alpha)^{\frac{\nu+2}{\nu}}\right) - \frac{C_{1\nu}}{r_s}\left((1+\alpha)^{\frac{\nu+1}{\nu}} + (1-\alpha)^{\frac{\nu+1}{\nu}}\right), \tag{22}$$

where the first term corresponds to the mean value of the kinetic energy contribution and the second term describes the exchange contribution. The energy $\frac{E(r_s)}{N}$ as a function of the mean distance $r_s$ has the extremum, i.e., $\frac{1}{N}\frac{\partial}{\partial r_s}E(r_s) = 0$, if the critical parameter $r_s*$ is equal to:

$$r_s* = \frac{2C_{2\nu}}{C_{1\nu}}\frac{\left((1+\alpha)^{\frac{\nu+2}{\nu}} + (1-\alpha)^{\frac{\nu+2}{\nu}}\right)}{\left((1+\alpha)^{\frac{\nu+1}{\nu}} + (1-\alpha)^{\frac{\nu+1}{\nu}}\right)}. \tag{23}$$

It is easy to see that if $r_s < r_s*$ the most preferable state is the state with the polarization degree $\alpha \to 0$, i.e., non-polarized; for the case $r_s > r_s*$ the energetically preferable state is polarized, with $\alpha \to 1$. The key role in the appearance of the spontaneous spin polarization is played by the coefficient relations $2C_{2\nu}/C_{1\nu}$, which is dependent on the space dimension. These relations for each space dimension, the same as for critical mean distances, are listed in Table 4. Analyzing the obtained results, it is possible to assume that the polarized state is realized in the 0D*-, 1D*- metallic cluster system (oscillator-trapped) and in the quantum-scaled 1D metallic system. The possibility of the polarized state arising for quasi-two dimensional systems is dependent on free-electron concentration and corresponds typically to semiconductors with $r_s > 2.356$ (very low concentration) for the quantum-scaled systems and $r_s > 1.191$ for the above-mentioned real experimental situation [6,7], considered in the oscillator-trapping model. That is why, despite the theoretical predictions of [2,8,13] the polarized state is observed in experiments [6,7,21,31]. Figure 1 shows the obtained dependence of the total mean value of the energy per electron (22) as a function of the $r_s$ parameter and the polarization degree $\alpha$ for the 3D-, 2D-, and 1D-cases.

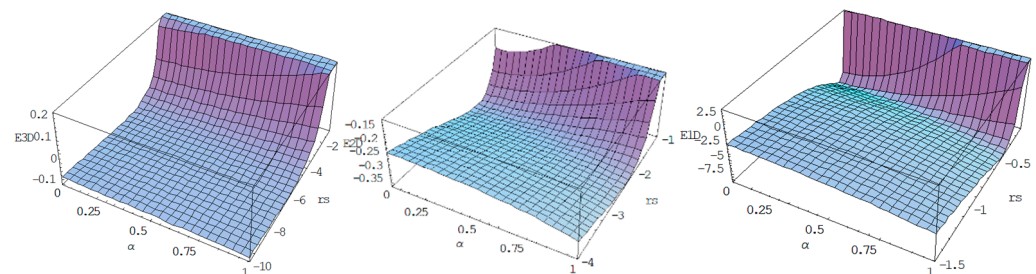

**Figure 1.** Dependences of the total mean value of the energy per electron E/N as a function of the $r_s$ parameter and the polarization degree $\alpha$ for the 3D-, 2D-, and 1D-cases, separately.

**Table 4.** Critical parameter.

| $\nu$ | $\frac{2C_{2\nu}}{C_{1\nu}}$ | $r_s* \to_{\alpha \to 1} = \frac{C_{2\nu}}{C_{1\nu}} 2^{\frac{2\nu+1}{\nu}}$ in a.u. |
|---|---|---|
| 1D $\nu = 1$ | 0.238 | 0.476 |
| 2D $\nu = 2$ | 1.666 | 2.356 |
| 3D $\nu = 3$ | 4.829 | 6.08 |
| 2D* $\nu = 4$ | $\frac{4}{3\pi^{1/4}} = 1.0015$ | 1.191 |
| 1D* $\nu = 5$ | $\sqrt{\pi} \cdot \left(\frac{\sqrt{2}}{15^3}\right)^{1/5} = 0.374131$ | 0.429763 |
| 0D* $\nu = 6$ | $\frac{6^{2/3}}{4\sqrt{\pi}} = 0.465728$ | $\frac{\left(6^2 \cdot \sqrt{2}\right)^{\frac{1}{3}}}{4\sqrt{\pi}} = 0.522762$ |

## 4. Conclusions

A system of electrons whose motion is free only in two (or one) spatial dimension and whose motion in the second and third dimensions corresponds to a discrete energy spectrum is called the dynamic two- (one)-dimensional (or quasi-two- (one)-dimensional) system. Thus, only two (or one) of the three components of the wave vector are good quantum numbers. We emphasize that such a system is not two- (one)-dimensional in the direct meaning, because the wave function of carriers is a function of three coordinates and electromagnetic fields propagate in three-dimensional space. It is virtually impossible to prepare such a sample with a really quantum second and third dimension size. The real experiments deal with quasi-two- (one)-dimensional clusters (for example, for a typical quantum well, width w~150 Å [23]), which is more conveniently described as a two- or one-dimensional oscillator system with the second (and third dimension) proportional to the oscillator's length. Spin aligning effects associated with the Coulomb exchange interaction of free electrons noticeably affect the magnetic ordering in 2D (1D) semiconductors and metals. This is because the main parameters characterizing the electron system in such a system in a magnetic field are quantities of the same order of magnitude, contrary to the 3D case. Now, it is important to underline that in this consideration of the low dimensional system with intermediate space-dimension we do not take into account very important corrections to the total energy of the system and the contributions of kinetic and exchange mean values to the total energy, like mass defects in the semiconductors and Fermi-liquid parameters. Their apparent mathematical definitions of an effective mass, $m^*$ and an effective fermi-liquid function $Z^*$, and their determinations in the 2D-electron system have drawn strong experimental and theoretical interest [4,23]. The self-energy is in itself a cumbersome problem still unsolved. An accurate determination of the self-energy and the renormalized mass valid for an unpolarized electron 2D-Fermi gas has been done in [27]. The main goal for our consideration is to understand how space dimension variation, in its pure form, directly affects the spin-polarization ability of a Fermi gas.

We should also make a remark concerning the Hartree-Fock method (HF) applied to a Fermi gas. It is well known that the neglect of correlations in the Hartree-Fock approximation introduces a strong bias towards spin polarization. Adding electron correlations stabilizes the unpolarized state in the 3D and 2D cases; more accurate quantum-Monte Carlo calculations indicate that spin-polarization occurs, if ever, in the liquid state of the homogeneous electron gas only in the very low density region close to the Wigner transition (see, for example, the works of Drummond and Needs [13,14,28]). The Landau Fermi-liquid method applied to the 2D Fermi system indicates a spin-polarization state and a paramagnetic–ferromagnetic phase transition in the lower electron density as pointed out by Hartree-Fock [4,21]. A direct comparison with experiments is problematic, since band structure effects cannot be neglected, see [7,15,16]. Indeed, Hartree-Fock's study is the least correct in describing the spin system, since it uses only the spin projections of each electron. Taking into account the commutation relations with the total spin value $[S^2, s_{iz}] \neq 0$, it is clear that we have the worst results, definitely, if the total spin projection is $Sz = 0$, due to the highest degeneration of the state in the total spin value. Therefore, we preferably use the exchange perturbation theory or the Landau Fermi-liquid method to describe spin systems with correlation effects (or spin ordering effects; see, for example, [4,29]). Nevertheless, we applied the HF method in this work to take into account the exchange interaction in the effects of spin ordering only because it provides the simplest way for analytical calculations and a possibility to compare the space dimension influence on the main parameters of the electronic system; or, in other words, a possibility to evaluate the influence of the spatial dimension on the relationship between the two main contributions, the exchange interaction and the kinetic energy, relative to the spin order. The HF method allowed us to make general calculations for all types of spatial dimensions. Another reason for the use of HF is the developed model of the oscillator trap, which takes into account the "quasi" character of a low-dimensional system. All calculations using various methods taking into account the effects of exchange correlation consider the low-dimensional system as a purely low-dimensional system and do not take into consideration the electron motion

in the quantized direction. In contrast to electronic systems, such motion is taken into account in low-dimensional Bose–Einstein condensates in the models of this oscillator, which is really useful for describing an electronic cluster system.

The main result of the presented work is the general expressions of mean energy per electron (22), (23) for an arbitrary space dimension with the analytically computed coefficients in them, as shown in Tables 2 and 3. We have estimated the effect of the space dimension on spin ordering. There are two competing factors: Non-force exchange, which affects the "interference" redistribution of the particle density in the spatial space (as a consequence of the Pauli exclusion principle); and the exchange Coulomb interaction, which is involved in the establishing of the equilibrium spin polarization. Thus, the principle of the indistinguishability of identical particles is manifested on two sides: It prevents spin polarization due to the kinetic energy of Fermi gas increases when spins are unpaired because free particles should occupy higher single-particle states; and, on the contrary, it establishes spin ordering thanks to the exchange interaction, and the degree of the ordering factor effect is dependent on the coefficient ratio. We have shown that the more the space dimension of the system decreases, the more the relative contribution of the exchange interaction increases as compared to the kinetic factor. This is in good agreement with the experimental results [15,16,24,30,31]. The metal–insulator transition experimentally observed in the metal–oxide-semiconductor field-effect transistors is characterized by a low concentration of carriers and has the spin nature of a metallic state [6,7,10]. The same happens in the 2D gas of current carriers in the Si–Ge and GaAs–GaAlAs heterojunctions [11,31]. It is impossible to achieve the polarized state of the degenerate electron gas for the 3D case. This statement is in agreement with [13,14].

**Author Contributions:** Conceptualization, E.O.; Methodology, E.O and F.O.; Software, F.O.; Validation, F.O., E.O.; Formal Analysis, E.O. and F.O.; Investigation, E.O.; Resources, E.O. and F.O.; Data Curation, F.O.; Writing-Original Draft Preparation, E.O. and F.O.; Writing-Review & Editing, E.O.; Visualization, F.O.; Supervision, E.O.; Project Administration, E.O.; Funding Acquisition, no.

**Funding:** This research received no external funding.

**Acknowledgments:** The authors are very thankful to the Organizing Committee of NICE Conference 2018 for the great scientific forum. In addition, one of the authors greatly appreciated the Peter the Great St. Petersburg Polytechnic University, its Program 5-100-2020, for the financial support for the participation in the NICE Conference 2018. We are very grateful to Nina V. Popova, Professor of the Department of Linguistics and Intercultural Communication, Institute of Humanities, Peter the Great St. Petersburg Polytechnic University, for the stylistic editing of the English version of our article.

**Conflicts of Interest:** There are no conflicts of interest to declare.

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
