# Peer review of "An Effect of the Space Dimension of Electron Fermi Gas upon the Spin Ordering in Clusters and Nanoparticles"

_chemosensors, doi:10.3390/chemosensors7010015_

Round 1
Reviewer 1 Report
The authors basically present a Hartree-Fock calculation for electrons in a confining harmonic oscillator potential to consider spontaneous spin ordering.
The presentation could be considerably improved:
a) there are many small language errors, e.g. line 28-30 verb is missing, line 33 -> systems, versions (both in plurial)
b) many definitions are missing, e.g. X in Eq.(1.1), main notations seem unconventional for me, e.g. "space dimensions nu=4,5,6", although not wrong, these small effects all together make the article hard to read
However, my main concern is the main conclusion drawn. It is well known that the neglect of correlations in the Hartree-Fock approximation introduces a strong bias towards spin polarization. Adding electron correlations stabilizes the unpolarized state, more accurate Quantum-Monte Carlo calculations indicate that spin-polarization, if ever, occurs in the liquid state of the homogeneous electron gas only in the very low density region close to the WIgner transition (see for example Drummond, Needs PRL 102, 126402 (2009). A direct comparison with experiments is problematic, since band structure effects cannot be neglected.
Therefore, in the present form, I cannot recommend publication.
Author Response
Dear Reviewer,
Reply to Reviewer 1.
First of all, we are grateful to the Reviewer for his / her careful and detailed review of our manuscript. All entries are helpful. In this regard, we make all the necessary corrections in the text.
1) The presentation could be considerably improved:
a) there are many small language errors, e.g. line 28-30 verb is missing, line 33 -> systems, versions (both in plural) – We corrected these sentences in accordance with the grammar of the English language. (Please see highlighted green).
b) many definitions are missing, e.g. X in Eq.(1.1), main notations seem unconventional for me, e.g. "space dimensions nu=4,5,6", although not wrong, these small effects all together make the article hard to read. We have added in the text the more detailed description of the universal variables “X” and the explanation of the mentioned space dimension.
“Here, space dimension ν=(α+ β)1,2,3 corresponds to real dimensions with α=1, (Px, ), α=1,2, (Px and Py ), or α=1,2,3, (Px,, Py,, Pz ) and β=0 , determining a free electron gas. Space dimensions ν= (α+ β)=4,5,6 corresponds to the α=3 (Px,, Py,, Pz ) and β=1, (Qx ) that gives ν=4 ; β=1,2,3 (Qx and Qy ), that gives ν=5; or β=1,2,3 (Qx , Qy,, Qz) . It determines a quasi-2 (with ν=4 ~ ν*=2) , quasi-1 (with ν=5 ~ ν*=1 ), or quasi-0 (with ν=6 ~ ν*=0) electron system. Because of the orthogonality and independence of all ν variables that determine the phase configuration we may use a universal variable set instead of variables. Then, we consider the Eq.(1.1) as an equation of spherical surface in ν –dimensional phase space:”
2) However, my main concern is the main conclusion drawn. It is well known that the neglect of correlations in the Hartree-Fock approximation introduces a strong bias towards spin polarization. Adding electron correlations stabilizes the unpolarized state, more accurate Quantum-Monte Carlo calculations indicate that spin-polarization, if ever, occurs in the liquid state of the homogeneous electron gas only in the very low density region close to the WIgner transition (see for example Drummond, Needs PRL 102, 126402 (2009). A direct comparison with experiments is problematic, since band structure effects cannot be neglected.)
As regards the remark concerning the Hartree-Fock method (HF) applied to a two-dimensional Fermi gas, the referee is objectively right. Indeed, Hartree-Fock is the least correct in describing the spin system, since it uses only the spin projections of each electron. Taking into account the commutation relations with the total spin value [S2 , siz]=/=0, it is clear that the worst result we have, namely, if the total spin projection is Sz=0, due to the highest degeneration of the state in total spin value. Therefore; We preferably use the exchange perturbation theory method or the Landau Fermi-liquid method to describe spin systems with correlation effects (or spin ordering effects). (See Example [4] and added to the list of references [28] by E. V. Orlenko and B. G. Matisov, Technical Physics, 2001, 46, (4), p. 463–465.). Nevertheless, we used the HF method in this work to take into account the exchange interaction in the effects of spin ordering only because of the simplest analytical calculations that make it possible to make comparisons between different values of spatial measurements. Or, in other words, to evaluate the influence of the spatial dimension on the relationship between the two main contributions, the exchange interaction and the kinetic energy, relative to the order of the spins. The HF method allowed us to perform general calculations for all types of spatial dimensions. Another reason for the use of HF is the developed model of the oscillator, which takes into account the "quasi" character of a low-dimensional system. All calculations using various methods that take into account the effects of exchange correlation consider the low-dimensional system as a purely low-dimensional system and do not take into account the motion of an electron in the quantized direction. In contrast to electronic systems, such motion is taken into account in low-dimensional Bose-Einstein condensates in the models of this oscillator, which is really useful for describing an electronic cluster system. In contrary to the 2-dimensional Fermi-system in the one-dimension case, the polarized state observed in different experiments (see for example Ref.[9]).
Because of importance of the Referee remark and its fundamental character, we included this discussion in the part “Conclusions”, if the Referee would agree.
We hope that after these corrections our manuscript would meet the high requirements of the Chemosensors journal.
Yours sincerely,
Elena Orlenko
Reviewer 2 Report
this article is written quite clearly. In principle, if authors will include in the introduction and/or conclusions sections of this manuscript some information about a practical application of their theoretical investigations, it will be very interesting for potential readers of Chemosensors journal. Some illustrative pictures or schemes will be also very useful. In my opinion, the article is written in good English (but I am not a native speaker, unfortunately). The manuscript contains relatively few literary references, which is quite typical for articles written by physicists or mathematicians.
I think that this manuscript deals with theoretical physical issues, but not with chemistry. However, probably, it could be useful for specialists in topic of scientific interests of authors. So, I think this work could be accepted for publication.
Author Response
Dear Reviewer,
We are very thankful to you for your kindness and helpful remarks. In accordance with these remarks, we added to the Introduction a few words about the possible application of the low-dimensional system under consideration for spin transistors:
“One of the possible applications of the low-dimensional electron Fermi-system is a so called spin transistor [32 S. Datta, B. Das, et al., Appl. Phys. Lett. 1990, 56 , 665]. The spin transistor is a magnetically-sensitive device made out of magnetic materials. It is currently a hot area of research and is an advanced version of the conventional electronic transistor. For functioning of this device first, the spins have to be injected from source into a non-magnetic layer and then transmitted to the collector. These non-magnetic layers are also called as semimetals, because they should have very large spin diffusion lengths. The operation of the spin transistor relies on the ability of electrons to store an information as a spin polarization. The band structure of the ferromagnetic emitter is such that an electric current is carried by electrons of one spin subband. It is spin polarized current and there is a current of magnetization. The emitter is viewed as a reservoir of particles (conducting electrons) that carry charge and magnetic moment. There is the rule: the emitter may exchange the electrons with the base (the nonmagnetic layer) only if those particles are the same “spin-polarization” as a base.”
The additional references also added in the list. Following the reviewer's comments, we presented the Figure, that shows the Dependences of the total mean-value of the energy per electron E/N as a function of the rs parameter and the polarization degree α for the 3D-, 2D- and 1D-cases, respectively.
We hope that after these corrections our manuscript would meet the high requirements of the Chemosensors journal.
Yours sincerely,
Elena Orlenko
Round 2
Reviewer 1 Report
The authors have taken into account my remarks and improved the paper.